# Novel Antidiabetic Agents and Their Effects on Lipid Profile: A Single Shot for Several Cardiovascular Targets

**DOI:** 10.3390/ijms241210164

**Published:** 2023-06-15

**Authors:** Francesco Piccirillo, Sara Mastroberardino, Annunziata Nusca, Lorenzo Frau, Lorenzo Guarino, Nicola Napoli, Gian Paolo Ussia, Francesco Grigioni

**Affiliations:** 1Fondazione Policlinico Universitario Campus Bio-Medico, Via Alvaro del Portillo, 200, 00128 Roma, Italy; f.piccirillo@unicampus.it (F.P.); sara.mastroberardino@unicampus.it (S.M.); lorenzo.frau@unicampus.it (L.F.); lorenzo.guarino@unicampus.it (L.G.); n.napoli@policlinicocampus.it (N.N.); g.ussia@policlinicocampus.it (G.P.U.); f.grigioni@policlinicocampus.it (F.G.); 2Research Unit of Cardiovascular Sciences, Department of Medicine and Surgery, Università Campus Bio-Medico di Roma, Via Alvaro del Portillo, 21, 00128 Roma, Italy; 3Research Unit of Endocrinology and Diabetes Department of Medicine and Surgery, Università Campus Bio-Medico di Roma, Via Alvaro del Portillo, 21, 00128 Roma, Italy

**Keywords:** cardiovascular diseases, diabetes, dyslipidemia, sodium glucose transporter-2 inhibitors (SGLT2i), dipeptidyl peptidase-4 inhibitors (DPP4i), glucagon-like peptide-1 (GLP-1) receptor agonists (GLP-1 RAs)

## Abstract

Type-2 diabetes mellitus (DM) represents one of the most important risk factors for cardiovascular diseases (CVD). Hyperglycemia and glycemic variability are not the only determinant of the increased cardiovascular (CV) risk in diabetic patients, as a frequent metabolic disorder associated with DM is dyslipidemia, characterized by hypertriglyceridemia, decreased high-density lipoprotein (HDL) cholesterol levels and a shift towards small dense low-density lipoprotein (LDL) cholesterol. This pathological alteration, also called diabetic dyslipidemia, represents a relevant factor which could promotes atherosclerosis and subsequently an increased CV morbidity and mortality. Recently, the introduction of novel antidiabetic agents, such as sodium glucose transporter-2 inhibitors (SGLT2i), dipeptidyl peptidase-4 inhibitors (DPP4i) and glucagon-like peptide-1 (GLP-1) receptor agonists (GLP-1 RAs), has been associated with a significant improvement in CV outcomes. Beyond their known action on glycemia, their positive effects on the CV system also seems to be related to an ameliorated lipidic profile. In this context, this narrative review summarizes the current knowledge regarding these novel anti-diabetic drugs and their effects on diabetic dyslipidemia, which could explain the provided global benefit to the cardiovascular system.

## 1. Introduction

Type-2 diabetes mellitus (DM) represents a well-known risk factor for cardiovascular diseases (CVD) [1]. Diabetic patients show a two to four times higher risk for CVD death than non-diabetic subjects [1], often showing a more aggressive atherosclerotic disease [2]. As stated in the latest ADA/EASD guidelines, type-2 DM is a chronic complex disease, and a multi-factorial management is needed to prevent or delay complications [3]. Interestingly, hyperglycemia and glycemic variability are not the only determinants of the increased cardiovascular risk in diabetic patients, as aggressive glycemic control did not significantly reduce major cardiovascular (CV) events [4,5,6]. Actually, a common metabolic alteration related to diabetes is dyslipidemia, characterized by a large range of lipid abnormalities which define so-called diabetic dyslipidemia, including hypertriglyceridemia, decreased high-density lipoprotein (HDL) cholesterol levels and a shift towards small dense low-density lipoprotein (LDL) cholesterol [7]. This pathological triad, frequently observed in patients with DM, albeit with optimal statin treatment, represents a relevant factor that promotes atherosclerosis and increases CV risk [8].

In recent years, novel anti-diabetic agents, including sodium-glucose cotransporter 2 inhibitors (SGLT2-i), glucagon-like peptide-1 (GLP-1) receptor agonists (GLP-1 RAs), and dipeptidyl peptidase-4 inhibitors (DPP4i), have been developed [9]. Beyond the glucose-lowering effect, achieved without significant hypoglycemia, these new molecules have represented a turning point in managing diabetic patients, since several studies highlighted their beneficial actions on CV morbidity and mortality [10]. Specifically, their benefit to the CV system seems partly related to an ameliorated lipid profile [11]. 

On these bases, this narrative review aims to summarize the current literature regarding novel anti-diabetic drugs and their effects on lipid profile, which could be responsible for the provided global benefit to the CV system.

Details regarding research methods are described in Appendix A.

## 2. Diabetic Dyslipidemia

Diabetic dyslipidemia is characterized by elevated levels of lipoproteins related to the development of atherosclerosis, involving very low-density lipoprotein (VLDL), small dense low-density lipoprotein (sdLDL), and chylomicrons [12]. In addition, raised levels of triglycerides (TG) and low HDL cholesterol levels are typically observed in diabetic patients [12]. Figure 1 schematically summarizes all lipid abnormalities observed in diabetic dyslipidemia. 

Notably, diabetic dyslipidemia is associated with insulin resistance and often occurs before the onset of overt diabetes [13]. Despite that in subjects with preserved insulin sensitivity, insulin inhibits VLDL hepatic production, in chronic insulin-resistance conditions the liver becomes resistant to the inhibitory insulin effects, thus maintaining a raised secretion of VLDL [14]. Furthermore, the activity of several tissue lipases involved in the regulation of lipoprotein levels is reduced in insulin-resistance states [15]. In addition, insulin resistance also causes reduced absorption of free fatty acids and enhanced lipolysis by adipocytes, inducing higher levels of serum free fatty acids and, subsequently, an overproduction of derived triglyceride-rich lipoproteins in the liver and bowels [15]. Diabetic dyslipidemia is also associated with an excess of carbohydrates, which are stored as glycogen in the liver and as triglycerides in adipose tissue [16]. When VLDL levels are increased, the plasma cholesteryl ester transfer protein (CETP) promotes the exchange of triglycerides in VLDL for cholesterol in HDL, inducing the formation of cholesterol-enriched VLDL particles and triglyceride-rich cholesterol-depleted HDL particles [17]. Indeed, this modification induces the production of highly atherogenic VLDL and less protective HDL particles, which are also quantitatively reduced [13]. Moreover, CETP could promote the transfer of triglycerides into LDL in exchange for LDL cholesteryl esters, thus inducing the formation of smaller and dense lipid-depleted LDL particles [13]. 

Finally, the net result of all these abnormalities is raised levels of VLDL, triglycerides and LDL associated with reduced HDL levels, which characterize the diabetic dyslipidemia, translating into an increased atherosclerotic risk in patients with DM. This scenario highlights the need for optimal treatment of these patients to target both glycemic and lipidic control.

## 3. Novel Anti-Diabetic Drugs

### 3.1. Sodium-Glucose Cotransporter 2 Inhibitors

#### 3.1.1. Mechanisms of Action and Pre-Clinical Evidence

Sodium-Glucose Cotransporter 2 inhibitors (SGLT2i), also called gliflozins, are a group of antidiabetic agents which act by inactivating the sodium-glucose cotransporter 2 (SGLT2) located in the brush border of the proximal renal tubules, thus inducing sodium and glucose excretion to increase glycemic control [18,19]. In order to avoid significant glucosuria and, thus, energy loss, SLGT1 cotransporters work by improving glucose reabsorption by up to 40%. The reabsorption limit is reached when the filtered glucose amounts to 300 mg/min/1.73 m^2^ (about 180–200 mg/dL) [20]. In a diabetic setting, blood glucose concentration, and therefore the amount filtered by kidneys, is higher, thus inducing an increased expression of sodium-glucose cotransporters in the renal tubule to improve reabsorption capacity [21]. In this regard, De Fronzo et al. showed that SGLT2i dapagliflozin ameliorates glycemic control by reducing the excretion threshold of glucose [22]. In addition, a better glycemic control induced by SGLT2i is reflected in a reduction in glycated hemoglobin (HbA1c) of about 0.5–1%, which leads to improved insulin sensitivity and raises pancreatic beta-cell function [23].

Beyond the known effects on glycemia, several studies showed how SGLT2i also induces substantial lipidic profile changes. These modifications are related to several chemical mechanisms (Figure 2), many of which are not yet fully understood [24].

Firstly, the produced glucosuria leads to a relative lack of glucose lacking and a fasting state, shifting energetic use to lipids through beta-oxidation [19]. This metabolic shift reduces cellular lipo-toxicity and oxidative stress, favoring ketone production and preventing myocardial damage and fibrosis [25]. In this regard, Briand et al. showed that empagliflozin induces an increased synthesis of ketones and fats by switching energetic metabolism from carbohydrate to lipid utilization and causes reduced intestinal cholesterol absorption [26]. Moreover, SGLT2i shows relevant effects on lipolysis and lipogenesis [27]. Empagliflozin administration inhibits fatty liver production, reduces hepatic lipogenesis, and induces lipolysis in a diabetic setting [28,29]. Additionally, Osatapahn et al. suggested that canagliflozin may trigger the same metabolic switch through increased production of fibroblast growth factor 21 (FGF21), which acts as a coordinator of fasting-induced metabolic pathways promoting lipolysis, and a decreased expression of genes involved in de novo lipogenesis [30]. Concordantly, canagliflozin has been demonstrated to stimulate activated adenosine monophosphate-activated protein kinase (AMPK), which in turn phosphorylates and activates acetyl-CoA carboxylase and reduces lipogenesis [31]. Several studies showed that SGLT2i could reduce lipid peroxidation due to its antioxidative effects [27]. Lipid peroxidation is a pathological event characterized by the oxidative degradation of lipids, which induces toxic metabolites and oxidative stress production. Therefore, in diabetic rats, canagliflozin and dapagliflozin were demonstrated to downregulate Nicotinamide adenine dinucleotide phosphate (NADPH) oxidase-dependent reactive oxygen species (ROS) production and malondialdehyde (MDA) levels in myocardial and renal cells [32,33]. Similarly, empagliflozin increased myocardial levels of sirtuin3 and superoxide dismutase in cardiac cells of diabetic mice, preventing oxidative stress and lipid peroxidation [34]. 

In addition, SGLT2i, by reducing glucose reabsorption, decreases systemic glucotoxicity and consequently improves insulin sensitivity, secretion, and effectiveness [35,36]. Therefore, the ameliorated insulin resistance leads to reduced liver synthesis and increased catabolism of triglyceride-rich lipoproteins [24]. Several studies suggested that SGLT2i could prevent the accumulation of free fatty acids in adipose tissue, favoring the oxidation and consequently utilization to provide energetic substrates [28]. Empagliflozin administration lowers glucose transporter type 4 (GLUT4) expression in abdominal adipose tissue, increasing lipid mobilization and reducing lipid accumulation to decrease glycerol formation in adipose tissue [28]. Similarly, in a study by Wallenius et al., dapagliflozin was demonstrated to increase fatty free acids’ mobilization and transport, rather than accumulation in the liver [37]. Specifically, these effects were reached with an increased clearance and flux of free fatty acids through raised oxidation [37]. Moreover, Herring et al. demonstrated how dapagliflozin could increase the oxidation of free fatty acids to produce ketones, thus inducing a metabolic switch, which increases hepatic ketogenesis [38]. The use of SGLT2 inhibitors is associated with reduction in visceral adipose tissue mass [39]. Specifically, treatment with empagliflozin for six weeks induced a decrease in fat mass, with the most evident reduction in perirenal adipose tissue, considered as a part of visceral adipose tissue [40]. In addition, the high urinary glucose loss induced by SGLT2i leads to a metabolic shift and increased fatty acids oxidation, thus preventing fat accumulation in adipose tissue and liver [41]. Moreover, the use of empagliflozin provides higher expression of genes involved in gluconeogenesis and reduced expression of genes involved in lipogenesis in liver, kidney and adipose tissue [40]. Furthermore, empagliflozin could also reduce hepatic fat content, decreasing insulin resistance and systemic inflammation [42]. Similarly, therapy with dapagliflozin in high-fat-fed mice could suppress lipid accumulation, especially in mesenteric adipose tissue, and may decrease the content of fatty acids in adipose tissue, but not in the liver [43]. Finally, empagliflozin administration attenuated weight gain by increasing metabolism and adipose tissue browning [44].

The effects of SGLT2i on cholesterol metabolism are debated, and different studies showed contrasting results [27]. The modifications in lipidic serum levels secondary to SGLT2i administration are firstly related to hemoconcentration induced by natriuresis [24]. In addition, LDL levels increase because of a decreased LDL receptor expression on the hepatocytes’ surface [24]. As is well known, the reduction in LDL receptors and the following diminution in liver lipid count induce a raised activity of hydroxy-methyl-glutaryl CoA (HMG-CoA) reductase, the main enzyme involved in cholesterol synthesis [45]. Briand et al. proved that HMG-CoA reductase activity was increased by 31% in empagliflozin-treated hamsters [26]. At the same time, an approximately 20% reduction in LDL receptor expression was observed [26]. These remarks suggest increased serum LDL levels via decreased hepatic catabolism after empagliflozin administration. Similarly, Basu et al. demonstrated that canagliflozin increases lipoprotein lipase activity and decreases post-prandial lipemia, raising serum LDL cholesterol levels in diabetic rats [46]. Specifically, the increased LDL cholesterol levels result from a reduced clearance of circulating LDL and a higher triglyceride-rich lipoproteins’ lipolysis [46]. Hence, Cha et al. observed that SGLT2i administration in diabetic patients induces an increase in HDL and LDL levels [47]. Finally, a recent systematic review and meta-analysis of randomized clinical trials showed that SGLT2i induces a significant increase in total cholesterol, LDL-cholesterol, and HDL-cholesterol serum levels, associated with decreased plasmatic triglycerides levels [45]. Otherwise, several studies demonstrated how SGLT2i administration could reduce circulating levels of cholesterol [27]. In this regard, in a study conducted by Osataphan et al. in diabetic mice, the use of canagliflozin was associated with a reduction in circulating cholesterol due to an inhibition of genes involved in its uptake and synthesis [30]. Concordantly, dapagliflozin was able to induce decreased serum levels of HDL cholesterol and triglycerides in diabetic patients [48]. In addition, other evidence did not show significant changes in the cholesterol profile after dapagliflozin and empagliflozin administration [49,50]. The effects on LDL cholesterol metabolism induced by SGLT2i are schematized in Figure 3.

#### 3.1.2. Clinical Evidence

The beneficial effects of SGLT-2i on cardiovascular morbidity and mortality in patients with a high cardiovascular risk are well established [51,52]. However, apart from the well-known benefits in treating diabetes and heart failure [53], new clinical effects of these drugs are emerging over time. Indeed, several studies suggested that SGLT2i could also modulate lipid metabolism, preventing or improving dyslipidemia independently from lowering-glucose effects [27]. Table 1 reports the main findings of studies investigating lipid effects of SGLT2 inhibitors.

In this regard, Calapkulu et al. evaluated the changes in the lipid profile after three and six months of dapagliflozin treatment and whether this was influenced by some variables (age, gender, diabetes duration, hypertension and concomitant use of insulin) [54]. The results showed a non-significant reduction in blood lipid levels after three months of treatment and a statistically significant reduction after six months, with no relevant effects of the investigated variables [54]. In a study by Bays et al., dapagliflozin administration caused a minor increase in LDL and HDL cholesterol levels compared to placebo, regardless of the presence or absence of baseline dyslipidemia. In contrast, the decrease in triglyceride levels was observed only in patients with a previous diagnosis of hyperlipidemia [55]. Other studies gave similar results, with an increase in total and LDL cholesterol levels and a reduction in triglycerides following the administration of dapagliflozin [56,57,58]. In addition, the EMPA-REG OUTCOME trial already showed the effects of empagliflozin in reducing HDL and increasing LDL cholesterol [59]. Furthermore, in patients with diabetes and hypertension, empagliflozin was associated with a significant reduction in blood pressure and glycated hemoglobin and a slight increase in total and LDL cholesterol. At the same time, no main changes in HDL or triglycerides were noted [60]. These results were confirmed by other studies, which showed increased serum levels of LDL in empagliflozin-treated patients [45,61]. Additionally, the CANVA(S) trial demonstrated a slight increase in HDL and LDL cholesterol in patients treated with canagliflozin compared with placebo [62]. In a randomized, double-blind, phase III study conducted by Bode et al., canagliflozin caused a spread in HDL cholesterol and LDL cholesterol and decreased triglycerides after 104 weeks of treatment [63]. Notably, canagliflozin showed the highest LDL cholesterol-raising potency compared to other SGLT2is, while empagliflozin was the one with the highest power to increase total cholesterol [45].

In conclusion, most studies conducted up to now showed increased LDL and HDL cholesterol levels (and, consequently, in total cholesterol) and decreased triglycerides following SGLT2i administration. The shady effects of SGLT2i on lipid metabolism could be responsible for their relative lower benefits regarding cardiovascular events compared to GLP1-RA. However, these findings were not statistically significant in any case. Further studies are needed to investigate the effect of these drugs on the lipid profile as an integral part of cardiovascular health.

### 3.2. Glucagon-like Peptide-1 Receptor Agonists 

#### 3.2.1. Mechanisms of Action and Pre-Clinical Evidence

Glucagon-like peptide-1 (GLP-1) is an incretin hormone secreted in the gut in response to food intake that increases insulin secretion, inhibits glucagon production, and acts on pancreatic B cells [64]. GLP-1 receptor agonists (GLP-1 RAs), existing in short and long-term formulations, improve hyperglycemia and delay gastric emptying. Long-acting GLP-1 RAs increase insulin production and suppress glucagon production, reducing postprandial glucose and fasting plasma glucose, while short-acting agonists reduce postprandial glucose mainly by slowing gastric emptying [65]. Beyond the well-known benefit on glycemic control, several studies showed the pleiotropic effects of GLP-1 RAs (Figure 4), consisting of anti-inflammatory effects and improvement in lipid profile and endothelial dysfunction, which protect against the development of atherosclerosis and cardiovascular disease [10,66]. 

Specifically, semaglutide and liraglutide inhibit atherosclerosis by modulating inflammatory pathways in low-density lipoprotein and apolipoprotein E receptor-deficient mice [67]. In apolipoprotein E–deficient mice (apoE−/−), Arakawa et al. showed that exendin-4 reduced the accumulation of monocytes/macrophages in the vascular wall at least in part by suppressing the inflammatory response in macrophages through the activation of the cAMP/PKA pathway [68]. Moreover, GLP-1 RAs prevent the progression of atherosclerotic lesions by inhibiting the formation of macrophage foam cell clusters and suppressing the expression of inflammatory cytokines (i.e., IL-1, IL-6, and TNF-α) [67,69]. GLP-1 RAs could also suppress foam cell formation by activating autophagy in oxidized-LDL monocytes [70] and reducing acyl-CoA:cholesterol acyltransferase 1 (ACAT1) expression and activity [71]. Interestingly, despite no significant change in absolute HDL cholesterol levels, liraglutide improved HDL properties on endothelium in mice [72], resulting in a raised nitric oxide (NO) bioavailability and anti-inflammatory effects [73]. In overweight/obese subjects with prediabetes, the addition of liraglutide to a calorie-restricted diet was also associated with a decrease in the higher density LDL cholesterol subclasses and a shift away from small LDL lipoproteins (phenotype B, associated with increased risk of coronary artery disease) towards larger, more buoyant LDL lipoproteins (phenotype A and AB, associated with reduced risk). Liraglutide administration also decreased total apo B concentration and the apo B/apo A-1 ratio [74]. Therefore, Osto et al. reported that, after roux-en-Y gastric bypass, elevated GLP-1 levels, through the activation of the GLP-1 receptor by liraglutide, activate an enzymatic cascade that leads to improved HDL-mediated NO production and endothelial antiapoptotic, antioxidant, and anti-inflammatory effects [72]. In addition, liraglutide favorably modulates lipid metabolism in enterocytes and hepatocytes, allowing a shift of HDL cholesterol toward smaller HDL particles, which showed a raised cardiovascular protective effect [17]. Furthermore, in high-fat-fed mice, the administration of GLP-1 RAs reduced VLDL cholesterol production and hepatic steatosis, other than better glycemic control [75]. Moreover, GLP-1 signaling reduces VLDL and triglycerides production by the liver, decreases hepatic triglycerides content by modulating key enzymes of lipid metabolism in the liver, and impairs hepatocyte de novo lipogenesis and β-oxidation [76]. GLP-1 RAs could also activate hepatic cyclic adenosine monophosphate (cAMP), resulting in the phosphorylation of cAMP-activated protein kinase (AMPK), which acts as a suppressor of lipogenesis [77]. A study conducted in diabetic rats showed how liraglutide could significantly decrease total cholesterol, triglycerides and LDL cholesterol by promoting the reversal of cholesterol transport in hepatocytes [78]. Finally, Feng Xu et al. showed that 4 weeks of exenatide treatment reduced body weight, improved lipid profile, and decreased serum levels of total cholesterol, triglycerides, and free fatty acids in diet-induced obese mice [79]. Previous studies gave controversial results regarding the effects of GLP-1 RAs treatment on lipolysis. Armstrong et al. demonstrated how the use of liraglutide was associated with a decrease in circulating free fatty acids levels in the fasting state and a raised insulin-mediated suppression of lipolysis [80]. Similarly, exenatide was able to reduce plasma free fatty acids during an oral glucose tolerance test [81]. Other studies conducted on diabetic rats confirmed these results, proving reduced levels of serum free fatty acids after GLP1-RAs administration [82,83]. Conversely, other findings showed minor or no significant impact on the levels of free fatty acids and glycerol after liraglutide treatment [84,85].

#### 3.2.2. Clinical Evidence

Increasing evidence confirmed the effects of GLP-1 RAs in decreasing plasmatic levels of total cholesterol, triglycerides and low-dense lipoprotein cholesterol in diabetic patients [86,87,88]. Liraglutide administration for over one year was associated with reduced body weight and improved metabolic control in patients with dyslipidemia or hypertension [89]. The amelioration in fasting lipid profile mainly resulted in a significant decrease in total cholesterol and plasma triglycerides and a rise in HDL cholesterol [89]. The use of GLP-1 RAs has also been related to improved basal and postprandial lipidic levels in several studies [74,90]. A large meta-analysis of 35 trials confirmed that GLP-1 RAs could induce reductions in LDL and total cholesterol, albeit that no significant increase in HDL cholesterol was observed [90]. Specifically, exenatide, liraglutide, and taspoglutide reduced total cholesterol with a range of −0.16 mmol/L (95% CI, −0.26 to −0.06) to −0.27 mmol/L (95% confidence interval [CI], −0.41 to −0.12) versus placebo and thiazolidinediones (range, −0.26 to −0.37 mmol/L) [90]. Moreover, a significant decrease in LDL cholesterol levels was detected for all GLP-1 RAs versus placebo (range, −0.08 to −0.16 mmol/L), thiazolidinediones (range, −0.16 to −0.24 mmol/L), and insulin (range, −0.10 to −0.19 mmol/L) [90]. Finally, exenatide, liraglutide, and taspoglutide decreased HDL cholesterol with a range of −0.06 mmol/L (95% CI, −0.11 to −0.01) to −0.13 mmol/L (95% CI, −0.17 to −0.10) compared with thiazolidinediones [90]. Concordantly, in the DURATION-6 open-label study conducted on 911 patients, treatment with liraglutide once daily or exenatide once weekly in diabetic patients induced a significant decrease in total, LDL, and non-HDL cholesterol and an increased HDL cholesterol at 26 weeks follow-up [91]. The LEAD-6 trial, conducted on 464 patients, showed that liraglutide reduced total cholesterol (0.20 mmol/L), LDL cholesterol (0.44 mmol/L), triglycerides (0.41 mmol/L), and HDL cholesterol (0.04 mmol/L) after 26 weeks, albeit a slight increase in very low-density lipoprotein cholesterol (0.2 mmol/L) [92]. Similar results were observed in patients treated with exenatide, allowing a decrease in total cholesterol (0.09 mmol/L), LDL cholesterol (0.40 mmol/L), triglycerides (0.23 mmol/L), and HDL cholesterol levels (0.05 mmol/L), associated with a rise in VLDL cholesterol (0.27 mmol/L) [92]. Finally, as shown in a meta-analysis by Liu et al. [93], liraglutide induced a greater reduction in blood lipid levels and body mass index than traditional therapies. 

Furthermore, administering high-dose semaglutide once a week was associated with improved lipid parameters [94,95]. Specifically, the use of semaglutide for 30 weeks significantly reduced total cholesterol, free fatty acids and LDL cholesterol levels compared with the placebo in the SUSTAIN-1 trial [94]. Moreover, treatment with semaglutide resulted in a higher decrease in triglycerides, LDL and total cholesterol values compared with insulin glargine at a 30-weeks follow-up, along with a positive effect on HbA1c and weight [95]. In addition, compared with exenatide, semaglutide induced a greater reduction in free fatty acids and triglycerides after 56 weeks of treatment [96]. Lastly, exenatide also shows an ameliorated lipid profile. Chiquette et al. showed that exenatide once a week had a positive effect on lipoprotein levels, apolipoprotein B (apo B) levels, and the apo B/apo A ratio [97]. In addition, triglycerides and VLDL cholesterol levels were significantly reduced with both once-weekly and twice-daily exenatide regimens [97]. Interestingly, Song et al. in a meta-analysis published in 2015 provided preliminary evidence in favor of GLP-1-based therapies in overcoming atherosclerosis development/progression, which is also substantiated by the significant decreases in the total cholesterol, LDL cholesterol and triglycerides [98]. Similar results were reported in a recent review of 57 trials, which demonstrated the beneficial effects of GLP-1 RAs on total and HDL cholesterol levels [99]. Interestingly, the combination treatment with GLP-1 RAs and SGLT2i significantly reduced LDL cholesterol levels in diabetic patients compared to monotherapy [100]. Moreover, in the PIONEER clinical trial program, patients treated with oral semaglutide experienced a reduction in LDL cholesterol levels compared to those receiving placebo, associated with improvements in HDL cholesterol and triglycerides levels [101]. The lipid-lowering effects of oral semaglutide include enhanced insulin secretion, reduced glucagon secretion, and decelerated gastric emptying [102]. Interestingly, the dual GLP-1 agonist and glucose-dependent insulinotropic peptide (GIP) agonist tirzepatide showed beneficial effects on lipid metabolism, improving insulin sensitivity and reducing lipoprotein biomarkers such as apolipoprotein C-III, apolipoprotein B, and large triglyceride-rich lipoprotein particles, as well as small low-density lipoprotein particles [103,104]. Furthermore, treatment with tirzepatide induced a reduction in metabolites associated with obesity [103]. Finally, as showed by Wilson and al., tirzepatide significantly reduced LDL cholesterol and non-HDL cholesterol levels compared with placebo [103].

Table 2 summarizes studies investigating lipid effects of GLP-1 RAs.

### 3.3. Dipeptidyl Peptidase-4 Inhibitors 

#### 3.3.1. Mechanisms of Action and Pre-Clinical Evidence

Dipeptidyl peptidase-4 inhibitors (DPP4i) inhibit the catalytic activity of dipeptidyl peptidase-4 (DPP4), thus preventing the degradation of the incretin hormones (i.e., GLP-1 and glucose-dependent insulinotropic peptide, GIP), and therefore stimulating postprandial insulin secretion and reducing hepatic glucose production through lowered glucagon secretion [105]. Indeed, DPP4i are approved as glucose-lowering drugs for diabetes mellitus treatment, with minimal risk of hypoglycemia, being well-tolerated in short- and long-term studies [10]. However, besides their action on glycemia, DPP4i could also induce benefits in lipid metabolism and reduce atherogenic development (Figure 5).

Hence, in hyperlipidemic mice, the administration of anagliptin significantly reduced the levels of total cholesterol and triglycerides, suppressing sterol regulatory element-binding protein activity in hepatocytes [106]. In addition, hepatic DPP4 leads to reduced glycogen storage, higher glucose output and raised lipid accumulation in the liver through an enhanced insulin resistance involving phosphorylation of insulin-receptor substrate 1 (IRS-1), mitogen-activated protein kinases (MAPK)/ERK, and protein kinase B/Akt [107]. Thus, all these pathways might be attenuated by the administration of DPP4i. In an in vitro study conducted by Mostafa et al., the incubation of adipocytes with vildagliptin induced an increased cholesterol efflux through the raised expression of the gene encoding for the adenosine triphosphate (ATP) binding cassette transporters family [108]. Moreover, several animal studies showed how DPP4i could reduce the accumulation of lipids and triglycerides in the liver [106,109] and kidneys [110]. Specifically, DDP4i could suppress the accumulation of triacylglycerol and diacylglycerol in hepatocytes by raising mitochondrial carbohydrate use and hepatic triacylglycerol secretion [109]. Similarly, the administration of teneligliptin reduced the lipid accumulation in the kidneys of apolipoprotein E knockout mice through the inhibition of renal lectin-like oxidized LDL receptor-1 [110]. Moreover, DPP4i could reduce fatty acids synthesis in the liver by upregulating the expression of carnitine palmitoyl-transferase-1 and increasing the activity of peroxisome proliferator-activated receptor-α and cyclic adenosine monophosphate (cAMP) reactive element binding homolog [111]. Interestingly, significant reduced levels of free fatty acids were observed after sitagliptin administration [112]. Similarly, treatment with DPP4i trelagliptin in rats induced decreased levels of free fatty acids secreted by fat cells, thus improving insulin resistance [113]. Kim and al. reported similar results in high-fat-fed animals, including reduction in plasma non-esterified fatty acids, after 14 weeks of treatment with evo-gliptin [114]. Moreover, a study conducted in cows showed how the use of DPP4i was related to reduced non-esterified fatty acids and triglycerides levels [115]. In addition, sitagliptin reduced intestinal cholesterol absorption in obese insulin-resistant mice, suggesting another potential mechanism translating into improved cholesterol metabolism [116]. Finally, Choi et al. showed how DPP4i could promote 5′ AMP-activated protein kinase activity, reducing fatty acid oxidation and suppressing oxidative stress and lipogenesis [117].

#### 3.3.2. Clinical Evidence

Several studies, summarized in Table 3, investigated the effects of DPP4i on the lipid profile in diabetic patients [118].

Specifically, this family of drugs has provided beneficial effects on total cholesterol and triglyceride levels [119]. Interestingly, most studies demonstrated that the positive effect on lipid metabolism in patients treated with DDP4i was mainly induced by the inhibition of cholesterol synthesis in hepatocytes, rather than the reduction of intestinal lipid transport [120,121,122]. Furthermore, available data suggest minor differences in lipid effects among the specific drugs, with vildagliptin appearing to be modestly superior to alogliptin and sitagliptin [123]. Remnant-like particle cholesterol (RLP-C) is an important coronary risk marker, and its levels are usually high in patients with chronic kidney disease (CKD) [124]. In these regards, the administration of DPP4i teneglipitin in diabetic patients with chronic kidney disease was significantly associated with reduced fasting glucose and RLP-C levels [124], suggesting a beneficial effect for the prevention and treatment of atherosclerosis in diabetic patients with CKD. In addition, DPP4 inhibition with sitagliptin could reduce the postprandial release of intestinal apoB-48-containing lipoproteins, preventing the accumulation of postprandial triglyceride-rich lipoprotein remnants, which play a key role in the development of atherosclerosis [125]. Moreover, treatment with anagliptin in diabetic patients leads to improved hyperlipidemia in both fasting and postprandial conditions, besides better glycemic control, favoring the prevention of cardiovascular disease [126]. Finally, a recent meta-analysis of 57 clinical trials investigated the effects on lipid parameters of novel antidiabetic agents [99]. Interestingly, DPP4i showed a significant impact in increasing HDL levels (weighted mean difference, WMD = −6.00 mg/dL, 95% CI: −8.43 to −3.57 mg/dL, *p* < 0.00001), while no significant changes were observed in total cholesterol, LDL and triglycerides levels [99].

## 4. Effects of Combination Therapy

Despite therapies’ development, glycemic control targets are not immediately reached, thus the combination therapy involving traditional hypoglycemic drugs such as metformin is widely spread. On these bases, Cheng and al. demonstrated that therapy with dapaglifozin plus metformin greater reduced the components of metabolic syndrome (body weight, BMI, waist circumference, fasting plasma glucose, triglycerides) in comparison to dapaglifozin or metformin monotherapies [127]. Moreover, dapaglifozin and metformin monotherapies were associated with similar modifications in HDL cholesterol and LDL cholesterol, while combination therapy provided greater increase in HDL cholesterol [127]. Similarly, dapagliflozin added to metformin induced a reduction in total body weight, mainly through reduction of fat mass and visceral adipose tissue [128]. In addition, triple therapy involving metformin, linaglipitin and dapagliflozin more effectively reduced LDL cholesterol rather than metformin monotherapy [129]. Moreover, treatment with metformin, SGLT2i and DPP4i improved total cholesterol and HDL cholesterol more than metformin monotherapy [129]. Furthermore, the addition of saxagliptin to metformin plus dapagliflozin produced a more effective decrease in total cholesterol and LDL cholesterol levels compared to therapy with metformin plus dapagliflozin [129]. Equally, dual therapy with exenatide plus dapagliflozin provided a greater reduction in triglycerides levels compared with dapagliflozin alone [130,131]. A recent review conducted by Li and al. showed how GLP-1 RAs and SGLT2i combination therapy led to higher reduction in body weight, body mass index and LDL cholesterol compared with monotherapy [100].

## 5. Novel Cholesterol-Lowering Drugs

Despite current recommendations, only 30% of patients with type-2 diabetes reach the LDL cholesterol targets [1,132]. Particularly, the primary reason of the failure to reach therapeutic goals with statin is poor adherence to therapy, mainly induced by the development of myalgia [133]. In recent years, the importance of reducing cardiovascular risk by controlling dyslipidemia has led to the development of novel cholesterol-lowering drugs, more effective than statins and ezetibime. Specifically, the use of proprotein convertase subtilisin/kexin type inhibitors (PCSK9i) provides about 50–60% reduction in LDL cholesterol [134]. Nevertheless, subcutaneous self-injection of these drugs is not acceptable for some patients. Inclisiran is a new PCSK9i that contemplates a biannual subcutaneous administration performed by healthcare professionals. A recent metanalysis of nine randomized clinical trials, showed how therapy with inclisiran was not associated with an increased incidence of new-onset diabetes [135]. Moreover, changes in LDL cholesterol induced by inclisiran did not result in significant impact on new-onset diabetes [135]. Similarly, Leiter and al. showed how treatment with inclisiran did not induce significant changes in glycated hemoglobin at 180 days follow-up [136]. Furthermore, the beneficial effects in lowering LDL cholesterol levels were reached regardless of the presence of diabetes [136].

Bempedoic acid is a novel drug, administered orally, which significantly reduces LDL cholesterol, triglycerides and apolipoprotein B and could induce positive effects on glucose metabolism and insulin sensitivity [137]. Specifically, through the activation of adenosine mono-phosphate-activated protein kinase (AMPK), it induces an inhibition of the hepatic production of glucose [138]. Gutierrez and al. showed how treatment with bempedoic acid in patients with type-2 DM did not result in worsening of glycemic parameters, including fasting plasma glucose concentration, postprandial measures and morning postprandial plasma glucose [139]. Furthermore, as showed by a recent trial, bempedoic acid significantly reduced glycated hemoglobin in patients with diabetes and pre-diabetes by −0.12% and −0.06%, respectively, and did not worsen fasting glucose compared to placebo [140]. In addition, bempedoic acid is associated with lower incidence of new-onset diabetes compared to placebo [140].

## 6. Conclusions

Cardiovascular diseases remain the leading cause of morbidity and mortality worldwide. One of the most important independent factors for the development of CVD is type-2 diabetes mellitus, commonly associated with lipid abnormalities which define “diabetic dyslipidemia”, a condition characterized by hypertriglyceridemia, low HDL-cholesterol, and high small dense LDL levels. Thus, it appears clear that effective management of diabetic dyslipidemia is relevant in reducing the risk of CVD in diabetic subjects. In this setting, novel anti-diabetic agents could represent remarkable game-changers for the global reduction of CV morbidity and mortality in diabetic patients because of their benefit on lipid metabolism beyond the glucose-lowering effects.

## Figures and Tables

**Figure 1 ijms-24-10164-f001:**
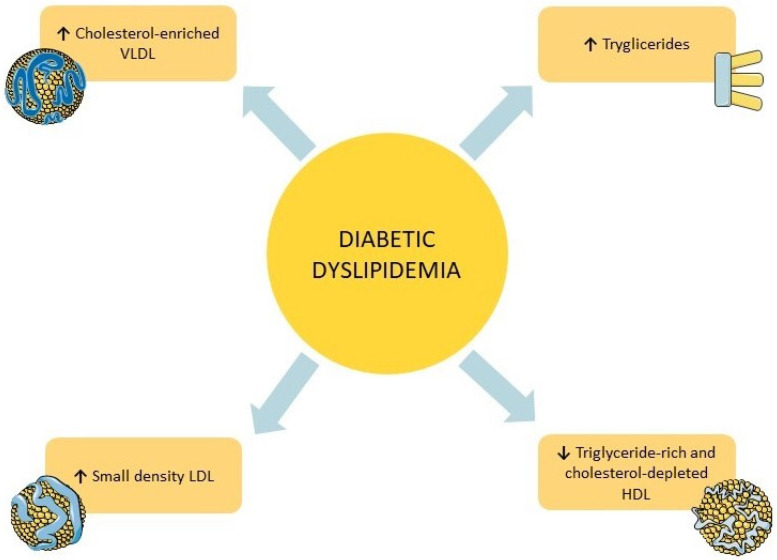
Quantitative and qualitative changes of lipid particles in diabetic dyslipidemia. HDL = high-density lipoprotein. LDL = low-density lipoprotein. VLDL = very low-density lipoprotein. ↑ = raised. ↓ = reduced.

**Figure 2 ijms-24-10164-f002:**
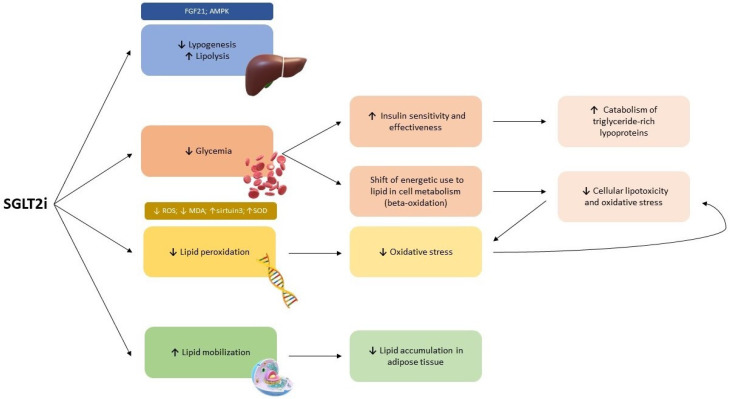
Modifications on lipid metabolism induced by SGLT2i. AMPK = adenosine monophosphate-activated protein kinase. FGF21 = fibroblast growth factor-21. MDA = malondialdehyde. ROS = reactive oxygen species. SOD = superoxide dismutase. SGLT2i = sodium glucose transporter 2 inhibitors. ↑ = raised. ↓ = reduced.

**Figure 3 ijms-24-10164-f003:**
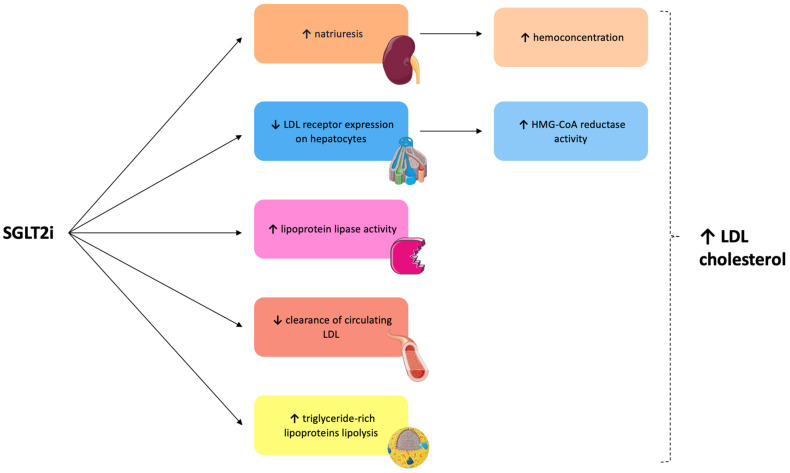
Effects on LDL cholesterol induced by SGLT2i. HMG-CoA = 3-hydroxy-3-methyl-glutaryl-coenzyme. LDL = low density lipoprotein; SGLT2i = sodium glucose transporter 2 inhibitors. ↑ = raised. ↓ = reduced.

**Figure 4 ijms-24-10164-f004:**
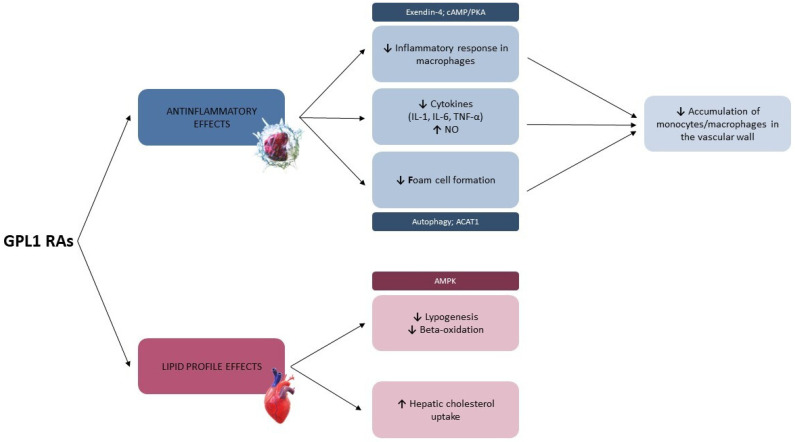
Modifications on lipid metabolism induced by GLP1 RAs. ACAT1 = acetyl-CoA acetyltransferase. AMPK = adenosine mono-phosphate-activated protein kinase. cAMP/PKA = cyclic adenosine monophosphate/protein kinase A. GLP1 RAs = glucagon-like peptide 1 receptor agonists. IL-1 = interleukin-1. IL-6 = interleukin-6. NO = nitric oxide. TNF-α = tumor necrosis factor α. ↑ = raised. ↓ = reduced.

**Figure 5 ijms-24-10164-f005:**
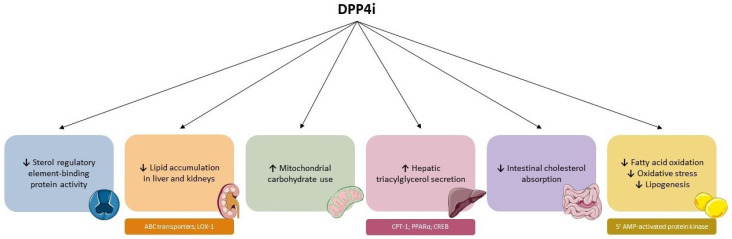
Modifications on lipid metabolism induced by DPP4i. ABC = ATP-binding cassette. AMP = adenosine monophosphate. CPT-1 = carnitine palmitoyltransferase-1. CREB = cAMP response element-binding protein. LOX-1 = lectin-type oxidized LDL receptor 1. PPARα = peroxisome proliferator-activated receptor–α. ↑ = raised. ↓ = reduced.

**Table 1 ijms-24-10164-t001:** Main results of studies investigating the effects of SGLT2i on lipid profile.

	Type of Study	Molecule	N° of Patients	Time of Follow-Up	Main Results
Calapkulu et al., 2019 [54]	Retrospective study	Dapaglifozin 10 mg	31 patients	3 and 6 months	At three months: - non-significant changes At six months: - ↓ LDL - (13.4 mg/dL) - ↓ total cholesterol (17.6 mg/dL) - ↓ triglyceride - (25.9 mg/dL)
Bays et al., 2017 [55]	Post-hoc analysis	Dapaglifozin 10 mg	4401 patients	6 months	↑ LDL ↑ HDL ↑ total cholesterol ↓ triglycerides
Matthaei et al., 2015 [56]	Phase 3b study	Dapaglifozin 10 mg	219 patients	6 months	↑ LDL ↑ HDL ↑ total cholesterol
Hayashi et al., 2017 [57]	Prospective study	Dapaglifozin 10 mg	80 patients	3 months	↑ LDL (0.5%) ↑ HDL (10.5%) ↑ total cholesterol (2.5%) ↓ triglycerides (12.4%)
Yanai et al., 2017 [58]	Retrospective study	Dapaglifozin 10 mg	249 patients (69 treated with dapaglifozin)	3 and 6 months	At three months: non-significant changes At six months: - non-significant changes in LDL, total cholesterol, and triglycerides - ↑ HDL
Zinman et al., 2015 [59]	Randomized controlled trial	Empaglifozin 10 or 25 mg	7028 patients	3 years	↑ LDL ↓ HDL
Tikkanen et al., 2015 [60]	Phase 3 study	Empaglifozin 10 or 25 mg	825 patients	14 weeks	With 10 mg: - non-significant changes With 25 mg: - ↑ LDL - ↓ total cholesterol - non-significant changes in HDL and triglycerides
Sánchez-García et al., 2020 [45]	Meta-analysis of 48 randomized controlled trials	Empaglifozin 10 or 25 mg AND Canagliflozin 100 or 300 mg	24,782 patients	Variable	↑ LDL ↑ HDL ↑ total cholesterol ↓ triglycerides
Liakos et al., 2014 [61]	Meta-analysis of 10 randomized controlled trials	Empaglifozin 10 or 25 mg	6203 patients	Variable	↑ LDL (4.5–6.5%)
Neal et al., 2014 [62]	Randomized controlled trial	Canaglifozin 100 or 300 mg	10,142 patients	Up to 78 weeks	↑ LDL ↑ HDL
Bode et al., 2015 [63]	Phase 3 study	Canaglifozin 100 or 300 mg	714 patients	104 weeks	↑ LDL ↑ HDL ↓ triglycerides

Abbreviations: HDL = high-density lipoprotein; LDL = low-density lipoprotein. ↑ = raised. ↓ = reduced.

**Table 2 ijms-24-10164-t002:** Main results of studies investigating the effects of GLP-1 RAs on lipid profile.

	Type of Study	Molecule	N° of Patients	Time of Follow-Up	Main Results
Ariel et al., 2014 [74]	Prospective study	Liraglutide	50 patients	14 weeks	↓ LDL ↓ non-HDL ↓ total cholesterol ↓ triglycerides ↓ apo-B ↓ apo-B/apo-A1 ratio shift of small LDL lipoproteins towards larger
Viswanathan et al., 2007 [86]	Retrospective study	Exenatide 5 mcg	52 patients	26 weeks	↓ total cholesterol ↓ triglycerides
Schwartz et al., 2010 [87]	Double-blinded, randomized, placebo-controlled study	Exenatide 10 mcg	35 patients	Up to 3 weeks	↓ RLP-cholesterol ↓ RLP triglycerides ↓ triglycerides ↓ apo-B48 ↓ apo-CIII
Hasegawa et al., 2018 [88]	Retrospective study	Various	317 patients	119 days	↓ LDL
Pi-Sunyer et al., 2015 [89]	Randomized controlled trial	Liraglutide 3.0 mg	3731 patients	56 weeks	↑ HDL ↓ total cholesterol ↓ triglycerides
Sun et al., 2015 [90]	Meta-analysis of 35 trials	Exenatide, Liraglutide, and Taspoglutide	14,340 patients	Al least 8 weeks	↓ LDL ↓ total cholesterol
Buse et al., 2009 [92] (LEAD-6)	Randomized controlled trial	Liraglutide and Exenatide	464 patients	26 weeks	↓ LDL ↓ HDL ↓ total cholesterol ↓ triglycerides ↑ VLDL
Buse et al., 2013 [91] (DURATION-6)	Randomized controlled trial	Liraglutide and Exenatide	911 patients (450 liraglutide, 461 exenatide)	26 weeks	↓ LDL ↑ HDL ↓ non-HDL ↓ total cholesterol
Liu et al., 2019 [93]	Meta-analysis of 13 trials	Liraglutide	1187 patients	At least 8 weeks	↓ triglycerides
Sorli et al., 2017 [94] (SUSTAIN 1)	Double-blind randomized trial	Semaglutide 0.5 mg Semaglutide 1.0 mg	388 patients	30 weeks	↓ LDL ↓ total cholesterol ↓ free fatty acids
Aroda et al., 2017 [95] (SUSTAIN 4)	Double-blind randomized trial	Semaglutide 0.5 mg Semaglutide 1.0 mg	1089 patients	30 weeks	↓ LDL ↓ total cholesterol ↓ triglycerides
Ahmann et al., 2018 [96] (SUSTAIN 3)	Double-blind randomized trial	Semaglutide 1.0 mg Exenatide	813 patients	56 weeks	(semaglutide > exenatide) ↓ triglycerides ↓ VLDL cholesterol ↓ free fatty acids
Chiquette et al., 2012 [97] (DURATION-1)	Post hoc analysis	Exenatide	211 patients	30 weeks	↓ triglycerides ↓ VLDL cholesterol ↓ apolipoprotein B (apoB) ↓ apo B/apo A ratio
Song et al., 2015 [98]	Meta-analysis of 31 trials	Various	Variable	Up to 52 weeks	↓ LDL ↓ total cholesterol ↓ triglycerides
Dar et al., 2022 [99]	Meta-analysis of 57 trials	Various	Variable	Between 12 weeks and 312 weeks	↑ HDL ↓ total cholesterol
Li et al., 2022 [100]	Meta-analysis of 8 trials	Various	1895 patients	At least 12 weeks	↓ LDL

Abbreviations: Apo = apolipoprotein; HDL = high-density lipoprotein; LDL = low-density lipoprotein; RLP = remnant-like particle; VLDL = very low-density lipoprotein. ↑ = raised. ↓ = reduced.

**Table 3 ijms-24-10164-t003:** Main results of studies investigating the effects of DPP4i on lipid profile.

	Type of Study	Molecule	N° of Patients	Time of Follow-Up	Main Results
Monami et al., 2012 [119]	Meta-analysis of 17 trials	Alogliptin, Dutogliptin, Linagliptin, Saxagliptin, Sitagliptin, Vildagliptin	Variable	Variable	↓ total cholesterol ↓ triglycerides
Ikegami et al., 2021 [120]	Single-arm trial	Anagliptin	14 patients	6 months	↓ LDL ↓ lathosterol
Nishida et al., 2020 [121]	Retrospective study	Sitagliptin, Vildagliptin, Teneligliptin, Alogliptin and Linagliptin	1809 patients	3 and 12 months	↓ HDL (in sitagliptin and vildagliptin users) ↓ total cholesterol and triglycerides (in sitagliptin, vildagliptin, and alogliptin users)
Kusunoki et al., 2016 [122]	Prospective study	Alogliptin 25 mg Sitagliptin 100 mg	129 patients 6 patients	6 months 12 months	↓ LDL ↓ total cholesterol ↓ triglycerides non-significant changes in LDL, HDL, and total cholesterol
Monami et al., 2012 [123]	Meta-analysis of 18 trials	Alogliptin, Dutogliptin, Linagliptin, Saxagliptin, Sitagliptin, Vildagliptin	Variable	Variable	↓ total cholesterol ↓ triglycerides vildagliptin > sitagliptin and alogliptin
Homma et al., 2017 [124]	Prospective study	Teneligliptin 20 mg	25 patients	12 months	↓ RLP cholesterol ↓ FPG
Tremblay et al., 2014 [125]	Randomized controlled trial	Sitagliptin 100 mg	22 patients	6 weeks	↓ triglycerides ↓ apoB-48 ↓ free fatty acids
Kakuda et al., 2015 [126]	Prospective study	Anagliptin 200 mg	-	12 weeks	↓ LDL ↓ non-HDL cholesterol ↓ RLP cholesterol ↓ total cholesterol ↓ triglycerides ↓ apoB-48
Dar et al., 2022 [99]	Meta-analysis of 57 trials	Variable	Variable	Variable	↑ HDL Non-significant changes in LDL, total cholesterol, and triglycerides

Abbreviations: apo = apolipoprotein; FPG = fasting plasma glucose; HDL = high-density lipoprotein; LDL = low-density lipoprotein; RLP = remnant like particle. ↑ = raised. ↓ = reduced.

## Data Availability

Not applicable.

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
