# Peer review of "Novel Antidiabetic Agents and Their Effects on Lipid Profile: A Single Shot for Several Cardiovascular Targets"

_ijms, 2023, doi:10.3390/ijms241210164_

Round 1
Reviewer 1 Report
I have analyzed the manuscript of Piccirillo and coauthors. In my opinion the presented review contains some interesting findings. However, I have some suggestions and critical points for the review.
1.Which type of diabetes treatment have authors described? It is not clarified in the manuscript.
2.I think that the very interesting point will be addition of the analysis of glucose-lowering effects of the new anti-atherogenic drugs such as inclisiran, bempedoic acid etc.
3.Authors should described methodology of the study in the separate subsection. Why authors did not include keyword "insulin resistance", "prediabetes" and "impaired glucose tolerance" in the study?
4.Which is the difference of gliflozins effects on adipose tissue and liver?
5.I think that the authors should analyze separately for each drug groups the effects on TAG metabolism, FA metabolism and glycerol metabolism.
Author Response
I have analyzed the manuscript of Piccirillo and coauthors. In my opinion the presented review contains some interesting findings. However, I have some suggestions and critical points for the review.
We appreciate the positive feedback from the reviewer, and we thank him/her for all the many helpful comments that we believe to have fully addressed in this revised version of the manuscript.
1.Which type of diabetes treatment have authors described? It is not clarified in the manuscript.
We thank the reviewer for the comment. We described type-2 diabetes, thus we specificized this in within the manuscript.
2.I think that the very interesting point will be addition of the analysis of glucose-lowering effects of the new anti-atherogenic drugs such as inclisiran, bempedoic acid etc.
We thank the reviewer for his/her suggestion. This point is very intriguing, as the effects of these novel drugs on glycemia are not fully understood and investigated. We added a short section entitled “Novel cholesterol-lowering drugs”, in which we tried to summarize current literature regarding glycemic parameters (Page 17, Lines 496 – 524).
3.Authors should described methodology of the study in the separate subsection. Why authors did not include keyword "insulin resistance", "prediabetes" and "impaired glucose tolerance" in the study?
We thank the reviewer for the comment. As journal’s guidelines does not provide a “methods” sections, we moved methodological description in “Appendix A” section (Page 18). Moreover, we did not use keywords suggested as diabetic dyslipidemia is usually present in diabetes mellitus and not in pre-diabetes condition. Similarly, the use of novel anti-diabetic drugs analyzed in our review is generally limited to overt diabetes.
4.Which is the difference of gliflozins effects on adipose tissue and liver?
We thank the reviewer for the comment. We analyzed different effects of gliflozins on adipose tissue in the specific section (Page 5, Lines 176 – 189).
5.I think that the authors should analyze separately for each drug groups the effects on TAG metabolism, FA metabolism and glycerol metabolism.
We thank the reviewer for his/her suggestion. We added more information about effects on lipids metabolism induced by different drugs in their respective sections (Page 5, Lines 165 – 176 / Page 10, Lines 319 – 327 / Page 14, Lines 424 – 434).
Reviewer 2 Report
Title: Novel Antidiabetic Agents and Their Effects on Lipid Profile: A Single Shot for Several Cardiovascular Targets
Authors: Francesco Piccirillo, Sara Mastroberardino, Annunziata Nusca, Lorenzo Frau, Lorenzo Guarino, Nicola Napoli, Gian Paolo Ussia, Francesco Grigioni
General remark:
The treatment of diabetes-related atherogenic dyslipidemia is as important a part of diabetes management as glycaemic control. In their paper, Francesco Piccirillo et al. reviewed the current literature regarding the effects of new hypoglycaemic drugs (SGLT2i, GLP1-RA, and DPPIVi) on the lipid profile. The concept of the paper is sound and the literature review is done decently, so I have only a few minor comments to include before the manuscript is accepted for publication.
Minor revisions:
1) Introduction
The control of dyslipidemia is part of a holistic model of diabetes care, according to the latest ADA/EASD guidelines As an introduction, it is worth referring to the assumptions of these guidelines
2) Section 3.1 Sodium-Glucose Cotransporter 2 Inhibitors
Figure 2 - please consider the effect of SGLT2i on LDL cholesterol. The figure shows only the beneficial effects of SGLT2i on lipid metabolism.
It is worth concluding this section with the observation that the equivocal effect of SGLT2i on lipids may determine its small (compared to GLP1-RA) protective effect on cardiovascular events, particularly stroke.
3) Section 3.2 Glucagon-like peptide-1 receptor agonists
Please provide data on the effectiveness of oral semaglutide in the modification of lipid profile.
Consider mentioning the impact of tirzepatide – a double GLP1 and GIP RA on lipids.
4) Treatment of diabetes is often based on polytherapy, so it is worth adding a section on the effects of combination therapies on the lipid profile, also taking into account 'old' hypoglycaemic drugs such as metformin.
The manuscript may benefit from the assistance of a native English speaker to correct some minor inaccuracies e.g. “This pathological alteration, also called diabetic dyslipidemia, represents a relevant factor which could promotes atherosclerosis and subsequently an increased CV morbidity and mortality.”
Author Response
General remark: The treatment of diabetes-related atherogenic dyslipidemia is as important a part of diabetes management as glycaemic control. In their paper, Francesco Piccirillo et al. reviewed the current literature regarding the effects of new hypoglycaemic drugs (SGLT2i, GLP1-RA, and DPPIVi) on the lipid profile. The concept of the paper is sound and the literature review is done decently, so I have only a few minor comments to include before the manuscript is accepted for publication.
We appreciate the positive feedback from the reviewer, and we thank him/her for all the many helpful comments that we believe to have fully addressed in this revised version of the manuscript.
Minor revisions:
1) Introduction
The control of dyslipidemia is part of a holistic model of diabetes care, according to the latest ADA/EASD guidelines As an introduction, it is worth referring to the assumptions of these guidelines.
We thank the reviewer the suggestion. We added this reference in the introduction section (Page 1, Lines 36 – 38).
2) Section 3.1 Sodium-Glucose Cotransporter 2 Inhibitors
Figure 2 - please consider the effect of SGLT2i on LDL cholesterol. The figure shows only the beneficial effects of SGLT2i on lipid metabolism.
We thank the reviewer for his/her suggestion.. We added a new figure (Figure 3) in which we specifically analysed the effects of SGLT2i on LDL cholesterol.
It is worth concluding this section with the observation that the equivocal effect of SGLT2i on lipids may determine its small (compared to GLP1-RA) protective effect on cardiovascular events, particularly stroke.
We thank the reviewer for the comment. We think that it’s a very intriguing point, and surely further studies are needed to better understand the effects of this relative new class of drugs on lipid metabolism and thus cardiovascular events.
3) Section 3.2 Glucagon-like peptide-1 receptor agonists
Please provide data on the effectiveness of oral semaglutide in the modification of lipid profile.
Consider mentioning the impact of tirzepatide – a double GLP1 and GIP RA on lipids.
We thank the reviewer for his/her suggestions. We added more information regarding effects of semaglutide and tirzepatide in the dedicated section (Page 11, Lines 379 – 390).
4) Treatment of diabetes is often based on polytherapy, so it is worth adding a section on the effects of combination therapies on the lipid profile, also taking into account 'old' hypoglycaemic drugs such as metformin.
We thank the reviewer for the comment. In this regard, we added a new section entitled “effects of combination therapy” (Page 16, Lines 472 – 481 / Page 17 Lines 482 – 493).
The manuscript may benefit from the assistance of a native English speaker to correct some minor inaccuracies e.g. “This pathological alteration, also called diabetic dyslipidemia, represents a relevant factor which could promotes atherosclerosis and subsequently an increased CV morbidity and mortality.”
We thank the reviewer for the comment. We did our best to improve the quality of manuscript.
Round 2
Reviewer 1 Report
Many thanks to authors for comprehensive response. I think that the manuscript can be accepted for publication.